# Whole Exome Sequencing in Drug-Induced Angioedema Caused by Angiotensin-Converting Enzyme Inhibitors: A Pilot Study in Five Patients

**DOI:** 10.3390/jcm14051659

**Published:** 2025-02-28

**Authors:** Alejandro Mendoza-Alvarez, Juan-Antonio Martinez-Tadeo, Eva Perez-Rodríguez, Javier Barrios-Recio, Jose-Carlos García-Robaina, Almudena Corrales, Itahisa Marcelino-Rodríguez, Jose-Miguel Lorenzo-Salazar, Rafaela González-Montelongo, Carlos Flores, Ariel Callero

**Affiliations:** 1Research Unit, Hospital Universitario Ntra. Señora de Candelaria, Instituto de Investigación Sanitaria de Canarias, 38010 Santa Cruz de Tenerife, Spain; amendoal@ull.edu.es (A.M.-A.); acorrales@fciisc.es (A.C.); imar-celi@ull.edu.es (I.M.-R.); cflores@ull.edu.es (C.F.); 2Allergy Service, Hospital Universitario Ntra. Señora de Candelaria, 38010 Santa Cruz de Tenerife, Spain; arkanum78@hotmail.com (J.-A.M.-T.); evaprod@hotmail.com (E.P.-R.); javierbarrios_17@hotmail.com (J.B.-R.); jcarlosgarciarobaina@gmail.com (J.-C.G.-R.); 3CIBER de Enfermedades Respiratorias (CIBERES), Instituto de Salud Carlos III, 28029 Madrid, Spain; 4Preventive Medicine and Public Health Area, Universidad de La Laguna, 38200 Santa Cruz de Tenerife, Spain; 5Instituto de Tecnologías Biomédicas (ITB), Universidad de La Laguna, 38200 Santa Cruz de Tenerife, Spain; 6Genomics Division, Instituto Tecnológico y de Energías Renovables, 38600 Santa Cruz de Tenerife, Spain; jmlorenzosalazar@iter.es (J.-M.L.-S.); rgonzalezmontelongo@iter.es (R.G.-M.); 7Facultad de Ciencias de la Salud, Universidad Fernando Pessoa Canarias, 35450 Las Palmas de Gran Canaria, Spain

**Keywords:** acquired angioedema, bradykinin, ACEi, NGS, exome

## Abstract

**Background and Objectives:** One of the most common causes of drug-induced angioedema (AE-DI) is related to reduced bradykinin breakdown after the use of certain medications. This is the case for forms of AE-DI due to the use of angiotensin-converting enzyme inhibitors (ACEi), which are used for the treatment of cardiovascular conditions. The causes of AE are not clear in these patients. Given the limited number of AE-ACEi genetic loci identified by genome-wide association studies, we opted to assess the utility of NGS of a panel of relevant genes to identify candidate genetic risk factors in severely affected patients. **Methods:** Five hypertensive patients from unrelated families with clinical AE-ACEi were included in the study. Whole-exome sequencing, variant calling, and annotation techniques were used. ANNOVAR v18.04.16 was used to annotate the variant calls. The resulting variants for each patient were assessed using the Hereditary Angioedema Database Annotation tool and Franklin genomic platform for variant prioritization and clinical impact interpretation. **Results:** The genetic variant rs6025 in the *F5* gene was identified in all recruited samples, which has been associated with an increase in blood clotting in AE-ACEi patients. In two patients, a common synonymous genetic variant of the *ACE* gene was found (rs4343). Finally, we identified the *ACE* genetic variant rs142947404 in only one patient. This variant has not been assessed in AE-ACEi. **Conclusions:** More studies will be needed to clarify the genetics involved in AE-DI. In this way, we will be able to try to predict future episodes of angioedema due to the use of ACEi.

## 1. Introduction

Drug-induced angioedema (AE-DI) is a medical condition often associated with reduced bradykinin (Bk) degradation following the use of specific medications [1]. A prominent example is AE-DI triggered by angiotensin-converting enzyme inhibitors (ACEi) used as a treatment for cardiovascular conditions, referred to as AE-ACEi. These medications interfere with the function of the angiotensin-converting enzyme, leading to an excessive accumulation of Bk in some patients [2]. AE-ACEi is uncommon, but it is calculated that the 12-month prevalence of AE-ACEi ranges from 0.004% to 0.026%. This percentage of affected patients varies between populations. Although the clinical manifestation of ACEi-AE is typically mild, there have been instances of fatal outcomes resulting from upper airway angioedema (AE) and subsequent obstruction [3]. Principal challenges in AE-ACEi patient management are related to differential diagnosis (AE-ACEi must be differentiated from other types of AE), diagnosis delay (influenced by the differential diagnosis), lack of specific treatment (management focuses primarily on discontinuation of ACEi intake), risk of lethal complications (mainly caused by upper airway obstruction), and genetic variability (ancestry and genetic factors influence in AE-ACEi susceptibility) [2].

Genetic testing in Bk-mediated AE (AE-BK) mostly focuses on rare hereditary forms, relying on conventional methods such as Sanger sequencing, multiplex ligation-dependent probe amplification (MLPA), and PCR. However, these genetic tests are rapidly evolving towards the use of next-generation sequencing (NGS), enabling highly efficient and holistic high-throughput genetic assessments. This transformation has paved the way for the identification of disease-causing variants, even in conditions with small genetic contributions. We have exposed the nuances of AE-ACEi, shedding light on the AE-DI forms, their connection to medication use, and the evolving methods for genetic testing, including whole exome sequencing (WES), as a powerful transformative tool to better understand and precisely manage this multifaceted condition in patients [2]. Given the limited number of AE-ACEi genetic loci identified by genome-wide association studies, we opted to assess the utility of NGS of a panel of relevant genes to identify candidate genetic risk factors in severely affected patients.

## 2. Materials and Methods

### 2.1. Patient Description

Five hypertensive patients from unrelated families with clinical AE-ACEi suspicion living in the Canary Islands (Spain) were recruited for this study (Table 1). Enalapril was the most frequently used drug among the patients in this study. They maintained treatment with ACEi for more than 15 months. The patients had at least one life-threatening episode of AE with dyspnea and oxygen saturation < 90%. The oropharynx was the most common location. All patients required assistance in the critical care unit. Antihistamines and corticosteroids were not effective. Complement studies were normal in all patients. The patients included in this study did not have any underlying pathology or chronic treatment that could influence the severity of AE-ACEi symptoms.

### 2.2. Sequencing, Identification, and Functional Annotation of Exome Variants

A total of 4 mL of peripheral blood was required for DNA extraction with Illustra^TM^ blood genomicPrep kit (GE Healthcare; Chicago, IL, USA). DNA concentration was assessed using the dsDNA BroadRange Assay Kit for the Qubit^®^ 3.0 Fluorometer (Thermo Fisher Scientific, Waltham, MA, USA) (Figure 1). DNA sequencing libraries were prepared using the DNA Prep with Exome 2.0 Plus Enrichment Kit (Illumina, San Francisco, CA, USA), with fragment sizes and concentrations assessed on a TapeStation 4200 (Agilent Technologies, Santa Clara, CA, USA) and sequencing obtained with a NovaSeq 6000 Sequencing System (Illumina, San Francisco, CA, USA) with paired-end 101-base reads. PhiX was loaded and sequenced at 1% as an internal control for the experiments.

The resulting sequencing reads were preprocessed with bcl2fastq v2.18 and mapped according to the hg19/GRCh37 reference genome with Burrows-Wheeler Aligner v0.7.15 [4]. Secondly, BAM files were processed with Qualimap v2.2.1 [5], SAMtools v1.3 [6], BEDTools [7], and Picard v2.10.10 (http://broadinstitute.github.io/picard, accessed on 22 September 2020) for quality control steps. Variant calling of small germline variants was performed using the Genome Analysis Toolkit (GATK) v.3.8 for the detection of nucleotide substitutions (SNVs) and small indels (<50 bp) following best practices [8]. The pipeline description is publicly available (https://github.com/genomicsITER/benchmarking/tree/master/WES, accessed on 17 January 2024).

Candidate “PASS” variants were filtered using SAMtools and VCFtools, combined with more restrictive sequencing metrics such as depth of coverage per position (≥20×), genotype quality (≥100), and mapping quality (≥50).

Identification of true candidate genetic variants causing AE-ACEi requires in-depth filtering strategies, such as gene biological function, gene location, allele frequency in reference populations, and possible disease relations contemplated on ClinVar [9], and The Human Gene Mutation Database [10]. ANNOVAR software [11] was used to incorporate this information in the previously filtered step, in combination with the pathogenicity scores including the Combined Annotation-Dependent Depletion (CADD) in the context of the Mutation Significance Cutoff (MSC), among others. In order to maximize the identification of candidate variants responsible for symptoms, the pathogenic potential of the contemplated variants was extracted from InterVar software (January 2018 release), according to the American College of Medical Genetics and Genomics (ACMG) guidelines [12].

This bioinformatic process for previously sequenced variants and the annotation of required information for filtering steps were performed using the Teide-HPC Supercomputing facility (http://teidehpc.iter.es/en, accessed on 17 January 2024).

### 2.3. Prioritization of Potential Deleterious Variants

The Hereditary Angioedema (HAE) Database Annotation tool (HADA, http://hada.hpc.iter.es/, accesed on 17 April 2024) was used as the first prioritization strategy for previously annotated variants [13]. This tool allows ruling out undiagnosed hereditary angioedema (HAE) patients, as recommended by the International WAO/EAACI guideline for the management of affected patients in its last recommendations update [14]. HADA easily matches and provides previously published information in the related scientific literature for submitted variants, performing a rapid identification of variants affecting the protein function involved in this heritable subtype of angioedema.

We used the Franklin genomic platform for variant prioritization and clinical impact interpretation as a second tier for identifying genetic factors that could be responsible for AE-ACEi symptoms [15]. For this approach, we extracted from the scientific literature the most common affected genes in AE-ACEi and designed a virtual gene panel composed of *ACE*, *BDKRB2*, *XPEPNP2*, *MME*, *F5*, *ETV6*, *DENND1B*, and *CRB1* [16]. This assessment was done by combining the search with a list of phenotypic abnormalities related to AE-BK (HP:0100665, HP:0025018, HP:0100540, HP:0002098, HP:0040315, HP:0031244, HP:0010783, HP:0002781, HP:0012027, HP:0011855).

### 2.4. Patient Population and Sequencing Summary

WES of the five patients provided an average of 6.58 Gb sequence, with an average of 100% of on-target reads and a median depth of 139.8×, and a transition/transversion ratio within the expected range (3.1 to 3.3).

## 3. Results and Discussion

AE is a relatively rare disease whose symptoms can be life-threatening. Due to the diverse etiology underlying symptom manifestation, it is key to identify the genetic risk factors using multiple strategies. In this sense, conducting a preliminary search in causal HAE genes is known to reduce the diagnostic odyssey of AE patients, as recommended by the International WAO/EAACI guideline (recommendation 3) [14]. The benefits of searching for variants in HAE causal genes have been proven by the identification of candidate variants with a protective effect against AE-ACEi [17]. In this first step, HADA did not reveal previously reported HAE causal variants present in recruited patients, reducing the possibility that the symptoms could be due to an undiagnosed HAE.

Secondly, sequencing data were also submitted to the Franklin platform to prioritize interesting likely deleterious variants responsible for inflammation symptoms (Table 2). Among the findings obtained by this tool, the variant c.1601C>T in the *F5* gene was identified in all recruited samples, which has been previously associated with an increase in blood clotting in affected patients using exome data [18]. A genetic meta-analysis of AE-ACEi found the association of this variant of the factor V Leiden at a genome-wide significance level, reinforcing the evidence of its pathogenic effect [3].

Franklin also prioritized the intronic variant c.989-53 T>G in the *CRB1* gene in all patients included in this study. This candidate variant has been associated with an increased risk of AE attacks because of ACEi intake in the ONTARGET dataset [16]. However, there is no further scientific information on this variant that provides additional evidence regarding the role of c.989-53 T>G in the pathogenic mechanism responsible for the symptoms.

The relevant function of the *ACE* gene in this disease makes the search for variants located in this gene a crucial step, allowing the identification of different candidate variants in the exome data. Moreover, ACE is one of the main components that inactivate the Bk activity. The synonymous c.2328 G>A genetic variant of *ACE* was found in two patients, AM_2498 and AM_2712. This variant was previously assessed to determine the association with AE-ACEi, specifically with captopril, and was associated with ACE activity [19]. In addition, the variant was reported as a possible risk factor for the development of AE-ACEi, likely due to its impact on the regulation of Bk levels and the ACE enzyme activity [20]. Although the c.2328 G>A in the *ACE* gene has been associated with AE-ACEi, the pathogenetic scores and population allele frequency of the variant support a benign classification. This scenario is common in clinical sequencing studies and highlights the intrinsic difficulties in the variant interpretation and its significance for the disease. The complexity lies in the fact that, by sequencing a larger number of genes, numerous variants are identified for which clinical relevance is uncertain, which complicates the prioritization, medical decision-making, and treatment prescription based on the data [21]. We also identified the *ACE* genetic variant c.3108 C>A in Patient AM_2450, which results in the change of asparagine to lysine (p.Asn1007Lys). This variant has not been assessed in AE-ACEi despite it being included in ClinVar as a variant of uncertain significance [22]. Some of the prediction scores such as SIFT and PROVEAN suggest a deleterious effect and its allele frequency in European populations is <0.1%. The findings on the *ACE* gene in this pilot study support the relevance of analyzing its variants to better understand the genetic predisposition to the disease.

In this study, significant limitations must be considered. The first constraint is the small sample size (five patients), which restricts the generalizability of the findings and limits the statistical power to establish strong associations between the identified genetic variants and ACEi-induced AE. A larger cohort will be necessary to confirm these preliminary observations and provide more robust conclusions. In this respect, this study should be considered a pilot study to evaluate alternative genetic screening strategies to identify novel genes governing AE-ACEi. Secondly, the study lacks functional data to validate the potential pathogenicity of the candidate variants. Without them, it is challenging to determine their biological impact and their role in the disease mechanism. This limitation also makes it difficult to exclude the involvement of certain variants and prioritize others that may emerge as stronger candidates when integrated with functional evidence. Future research incorporating larger patient cohorts and functional validation studies will be essential to refine the interpretation of genetic findings and enhance their clinical applicability. Among the strengths, it represents an innovative approach by using exome data to explore the genetic basis of AE-ACEi, expanding the identification of causal genetic variation and opening new research horizons beyond typical genetic screenings. Our findings are supported by evidence from the scientific literature and genetic databases, reinforcing the validity of the identified variants and their potential involvement in the pathogenesis of AE-ACEi. In addition, the use of advanced bioinformatics tools and variant prioritization strategies has allowed the identification of candidate genetic variants with previous evidence of association with AE-ACEi, suggesting that these results could be relevant for future studies with larger cohorts. Finally, our results also support the importance of further investigating variants in the *ACE* gene and other genes of the renin–angiotensin axis, which could contribute to the development of more precise strategies for the prevention and management of ACEI-induced AE.

In conclusion, we have explored the role of genetic variants in susceptibility to AE-ACEi by analyzing exome data from severely affected patients. Despite the limited sample size, we have uncovered the next steps for further genetic screening of AE-ACEi patients to improve the risk prediction and potentially prevent future episodes of AE induced by this drug.

**Table 2 jcm-14-01659-t002:** Sequencing metrics of candidate variants prioritized by Franklin in exome data of patients with AE-ACEi using the virtual gene panel.

Gene	Individual ID	Total Depth (ref/alt)	HGVS Coding	HGVS Protein	Effect	ACMG Class	ACMG Criteria	gnomAD	CADD Phred Score	MSC 95% HGMD	Predicted Effect [Reference]
*F5*	AM2419	156 (0/156)	NM_000130.5:c.1601 C>T	p.Gln534Arg	Missense	Likely pathogenic	PM1 (moderate), PM5 (moderate), PM2 (supporting)	Absent	0.255	12.286	Increased blood clotting [17,18]
	AM2450	124 (0/124)									
	AM2498	145 (75/70)									
	AM2569	121 (0/121)									
	AM2712	125 (0/125)									
*CRB1*	AM2419	101 (0/101)	NM_201253.3:c.989-53 T>G	none (intronic)	Regulatory (intronic)	Benign	BA1 (stand-alone), BP4 (strong), BP6 (moderate)	0.7967	0.446	11.653	Higher risk of ACEi induced angioedema [15]
	AM2450	117 (0/117)									
	AM2498	96 (1/95)									
	AM2569	87 (47/40)									
	AM2712	102 (0/102)									
*ACE*	AM2498	213 (138/75)	NM_000789.4:c.2328 G>A	p.Thr776=	Synonymous	Benign	BA1 (stand alone), BP6 (very strong), BP4 (strong), BP7 (supporting)	0.4727	0.083	0.177	Increased ACE activity [19,20]
	AM2712	206 (136/70)									
*ACE*	AM2450	88 (39/49)	NM_000789.4:c.3108 C>A	p.Asn1036Lys	Missense	Uncertain significance	BP4 (strong), BP1 (supporting), PM2 (supporting)	9.6 × 10^−4^	22.7	0.177	Not reported in the scientific literature [22]

ACMG: American College of Medical Genetics and Genomics; CADD: Combined Annotation Dependent Depletion; gnomAD: Genome Aggregation Database; HGMD: Human Gene Mutation Database; HGVS: Human Genome Variation Society; MSC: Mutation Significance Cutoff.

## Figures and Tables

**Figure 1 jcm-14-01659-f001:**
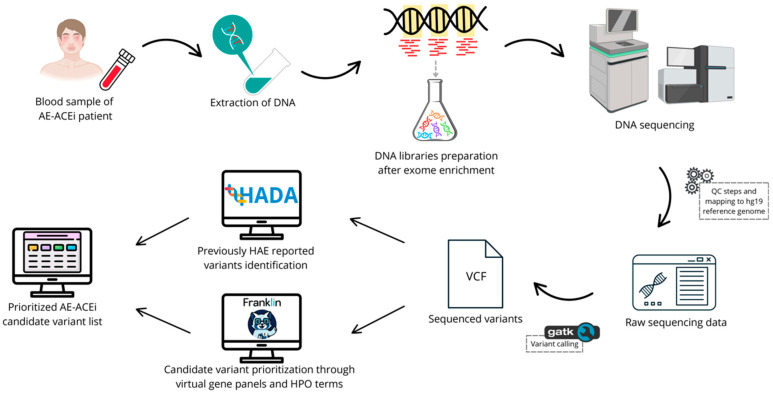
Overview of DNA sequencing and data processing for AE-ACEi candidate genetic variant prioritization.

**Table 1 jcm-14-01659-t001:** Demographics and clinical data of patients.

ID	Sex	Age	C4 mg/dL	C1q mg/dL	Drug	HTA Diagnosis	Onset of ACEI	Onset of AE (Months)	Localization
AM_2450	Male	64	25.5	18	Enalapril	2014	2014	47	Oropharynx
AM_2419	Male	43	26.1	19	Enalapril	2016	2016	27	Oropharynx
AM_2498	Male	62	24.3	21	Enalapril	2010	2010	82	Oropharynx, lips
AM_2569	Male	49	28.0	17	Ramipril	2019	2020	16	Oropharynx, lips
AM_2712	Female	90	27.2	ND	Perindopril	2010	2012	78	Oropharynx

ID: identification number; ND: no data; ACEI: angiotensin-converting enzyme inhibitors; AE: angioedema.

## Data Availability

The data presented in this study are available on request from the corresponding author. The data are not publicly available due to the restrictions established by the HUNSC Ethics Committee (CHUNSC_2020_95, 26 November 2020).

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
