# Peer review of "Whole Exome Sequencing in Drug-Induced Angioedema Caused by Angiotensin-Converting Enzyme Inhibitors: A Pilot Study in Five Patients"

_jcm, 2025, doi:10.3390/jcm14051659_

Round 1

Reviewer 1 Report

Comments and Suggestions for Authors

Dear author, The paper is interesting, but there are some mistakes in the nomenclature of angioedema caused by ACE-inhibitor use.

Please adequate the nomenclature to the newer classification inserted in the paper by Resheef et al. that needs to be quoted: “Reshef A, Buttgereit T, Betschel SD, Caballero T, Farkas H, Grumach AS, et al. Definition, acronyms, nomenclature, and classification of angioedema (DANCE): AAAAI, ACAAI, ACARE, and APAAACI DANCE consensus. J Allergy Clin Immunol. (2024) 154:398–411.e1. doi: 10.1016/j.jaci.2024.03.024”

Quote number 12 is not the latest version of the Guidelines: “Maurer M, Magerl M, Betschel S, Aberer W, Ansotegui IJ, Aygören-Pürsün E, Banerji A, Bara NA, Boccon-Gibod I, Bork K, Bouillet L, Boysen HB, Brodszki N, Busse PJ, Bygum A, Caballero T, Cancian M, Castaldo A, Cohn DM, Csuka D, Farkas H, Gompels M, Gower R, Grumach AS, Guidos-Fogelbach G, Hide M, Kang HR, Kaplan AP, Katelaris C, Kiani-Alikhan S, Lei WT, Lockey R, Longhurst H, Lumry WR, MacGinnitie A, Malbran A, Martinez Saguer I, Matta JJ, Nast A, Nguyen D, Nieto-Martinez SA, Pawankar R, Peter J, Porebski G, Prior N, Reshef A, Riedl M, Ritchie B, Rafique Sheikh F, Smith WB, Spaeth PJ, Stobiecki M, Toubi E, Varga LA, Weller K, Zanichelli A, Zhi Y, Zuraw B, Craig T. The international WAO/EAACI guideline for the management of hereditary angioedema-The 2021 revision and update. Allergy. 2022 Jul;77(7):1961-1990. doi: 10.1111/all.15214. Epub 2022 Feb 3. PMID: 35006617.

When searching PubMed with three different keywords: angioedema, angiotensin-converting enzyme, and genetic, many new papers were published. Please enrich the result and discussion section with these new papers:

https://pubmed.ncbi.nlm.nih.gov/?term=%28%28angioedema%29+AND+%28angiotensin-converting+enzyme%29%29+AND+%28genetic%29&sort=date

For example:

·        Mugo JW, Day C, Choudhury A, Deetlefs M, Freercks R, Geraty S, Panieri A, Cotchbos C, Ribeiro M, Engelbrecht A, Lisa K Micklesfield, Ramsay M, Sarah P, Peter J. A GWAS of ACE Inhibitor-Induced Angioedema in a South African Population. medRxiv [Preprint]. 2024 Sep 15:2024.09.13.24313664. doi: 10.1101/2024.09.13.24313664. PMID: 39314982; PMCID: PMC11419215.

·        Mathey CM, Maj C, Eriksson N, Krebs K, Westmeier J, David FS, Koromina M, Scheer AB, Szabo N, Wedi B, Wieczorek D, Amann PM, Löffler H, Koch L, Schöffl C, Dickel H, Ganjuur N, Hornung T, Buhl T, Greve J, Wurpts G, Aygören-Pürsün E, Steffens M, Herms S, Heilmann-Heimbach S, Hoffmann P, Schmidt B, Mavarani L, Andresen T, Sørensen SB, Andersen V, Vogel U, Landén M, Bulik CM; Estonian Biobank Research Team; DBDS Genomic Consortium; Bygum A, Magnusson PKE, von Buchwald C, Hallberg P, Rye Ostrowski S, Sørensen E, Pedersen OB, Ullum H, Erikstrup C, Bundgaard H, Milani L, Rasmussen ER, Wadelius M, Ghouse J, Sachs B, Nöthen MM, Forstner AJ. Meta-analysis of ACE inhibitor-induced angioedema identifies novel risk locus. J Allergy Clin Immunol. 2024 Apr;153(4):1073-1082. doi: 10.1016/j.jaci.2023.11.921. Epub 2024 Jan 31. PMID: 38300190.

·        Naik H, O'Connor MY, Sanderson SC, Pinnell N, Dong M, Wiegand A, Obeng AO, Abul-Husn NS, Scott SA. Pharmacogenomic knowledge and awareness among diverse patients treated with angiotensin converting enzyme inhibitors. Pharmacogenomics. 2023 Dec;24(18):921-930. doi: 10.2217/pgs-2023-0191. Epub 2023 Dec 6. PMID: 38054855; PMCID: PMC10794943.

·        Mathey CM, Maj C, Scheer AB, Fazaal J, Wedi B, Wieczorek D, Amann PM, Löffler H, Koch L, Schöffl C, Dickel H, Ganjuur N, Hornung T, Forkel S, Greve J, Wurpts G, Hallberg P, Bygum A, Von Buchwald C, Karawajczyk M, Steffens M, Stingl J, Hoffmann P, Heilmann-Heimbach S, Mangold E, Ludwig KU, Rasmussen ER, Wadelius M, Sachs B, Nöthen MM, Forstner AJ. Molecular Genetic Screening in Patients With ACE Inhibitor/Angiotensin Receptor Blocker-Induced Angioedema to Explore the Role of Hereditary Angioedema Genes. Front Genet. 2022 Jul 18;13:914376. doi: 10.3389/fgene.2022.914376. PMID: 35923707; PMCID: PMC9339951.

Author Response

Dear author, the paper is interesting, but there are some mistakes in the nomenclature of angioedema caused by ACE-inhibitor use. Please adequate the nomenclature to the newer classification inserted in the paper by Resheef et al. that needs to be quoted: “Reshef A, Buttgereit T, Betschel SD, Caballero T, Farkas H, Grumach AS, et al. Definition, acronyms, nomenclature, and classification of angioedema (DANCE): AAAAI, ACAAI, ACARE, and APAAACI DANCE consensus. J Allergy Clin Immunol. (2024) 154:398–411.e1. doi: 10.1016/j.jaci.2024.03.024”

Thank you very much for taking your time to revise our manuscript. Thank you for the compliments and the insights. We agree with the reviewer about the incorrect nomenclature indicated in the manuscript and we have updated them in the revised version according to Reshef et al., 2024 guidelines. The title of the revised manuscript has been modified to “Whole exome sequencing in drug induced angioedema by angiotensin-converting enzyme inhibitors: a pilot study in five patients”. Reshef et al., 2024 citation has been incorporated to the reference list in the revised version of the manuscript (lines 246-248).

Quote number 12 is not the latest version of the Guidelines: “Maurer M, Magerl M, Betschel S, Aberer W, Ansotegui IJ, Aygören-Pürsün E, Banerji A, Bara NA, Boccon-Gibod I, Bork K, Bouillet L, Boysen HB, Brodszki N, Busse PJ, Bygum A, Caballero T, Cancian M, Castaldo A, Cohn DM, Csuka D, Farkas H, Gompels M, Gower R, Grumach AS, Guidos-Fogelbach G, Hide M, Kang HR, Kaplan AP, Katelaris C, Kiani-Alikhan S, Lei WT, Lockey R, Longhurst H, Lumry WR, MacGinnitie A, Malbran A, Martinez Saguer I, Matta JJ, Nast A, Nguyen D, Nieto-Martinez SA, Pawankar R, Peter J, Porebski G, Prior N, Reshef A, Riedl M, Ritchie B, Rafique Sheikh F, Smith WB, Spaeth PJ, Stobiecki M, Toubi E, Varga LA, Weller K, Zanichelli A, Zhi Y, Zuraw B, Craig T. The international WAO/EAACI guideline for the management of hereditary angioedema-The 2021 revision and update. Allergy. 2022 Jul;77(7):1961-1990. doi: 10.1111/all.15214. Epub 2022 Feb 3. PMID: 35006617.”

Thank you very much for rising this point. We have indicated the latest version of the international WAO/EAACI guidelines in the revised version of the manuscript as is indicated by the reviewer (lines 275-276).

When searching PubMed with three different keywords: angioedema, angiotensin-converting enzyme, and genetic, many new papers were published. Please enrich the result and discussion section with these new papers:

https://pubmed.ncbi.nlm.nih.gov/?term=%28%28angioedema%29+AND+%28angiotensin-converting+enzyme%29%29+AND+%28genetic%29&sort=date

For example:

    •  

Mugo JW, Day C, Choudhury A, Deetlefs M, Freercks R, Geraty S, Panieri A, Cotchbos C, Ribeiro M, Engelbrecht A, Lisa K Micklesfield, Ramsay M, Sarah P, Peter J. A GWAS of ACE Inhibitor-Induced Angioedema in a South African Population. medRxiv [Preprint]. 2024 Sep 15:2024.09.13.24313664. doi: 10.1101/2024.09.13.24313664. PMID: 39314982; PMCID: PMC11419215.

    •  

Mathey CM, Maj C, Eriksson N, Krebs K, Westmeier J, David FS, Koromina M, Scheer AB, Szabo N, Wedi B, Wieczorek D, Amann PM, Löffler H, Koch L, Schöffl C, Dickel H, Ganjuur N, Hornung T, Buhl T, Greve J, Wurpts G, Aygören-Pürsün E, Steffens M, Herms S, Heilmann-Heimbach S, Hoffmann P, Schmidt B, Mavarani L, Andresen T, Sørensen SB, Andersen V, Vogel U, Landén M, Bulik CM; Estonian Biobank Research Team; DBDS Genomic Consortium; Bygum A, Magnusson PKE, von Buchwald C, Hallberg P, Rye Ostrowski S, Sørensen E, Pedersen OB, Ullum H, Erikstrup C, Bundgaard H, Milani L, Rasmussen ER, Wadelius M, Ghouse J, Sachs B, Nöthen MM, Forstner AJ. Meta-analysis of ACE inhibitor-induced angioedema identifies novel risk locus. J Allergy Clin Immunol. 2024 Apr;153(4):1073-1082. doi: 10.1016/j.jaci.2023.11.921. Epub 2024 Jan 31. PMID: 38300190.

    •  

Naik H, O'Connor MY, Sanderson SC, Pinnell N, Dong M, Wiegand A, Obeng AO, Abul-Husn NS, Scott SA. Pharmacogenomic knowledge and awareness among diverse patients treated with angiotensin converting enzyme inhibitors. Pharmacogenomics. 2023 Dec;24(18):921-930. doi: 10.2217/pgs-2023-0191. Epub 2023 Dec 6. PMID: 38054855; PMCID: PMC10794943.

    •  

Mathey CM, Maj C, Scheer AB, Fazaal J, Wedi B, Wieczorek D, Amann PM, Löffler H, Koch L, Schöffl C, Dickel H, Ganjuur N, Hornung T, Forkel S, Greve J, Wurpts G, Hallberg P, Bygum A, Von Buchwald C, Karawajczyk M, Steffens M, Stingl J, Hoffmann P, Heilmann-Heimbach S, Mangold E, Ludwig KU, Rasmussen ER, Wadelius M, Sachs B, Nöthen MM, Forstner AJ. Molecular Genetic Screening in Patients With ACE Inhibitor/Angiotensin Receptor Blocker-Induced Angioedema to Explore the Role of Hereditary Angioedema Genes. Front Genet. 2022 Jul 18;13:914376. doi: 10.3389/fgene.2022.914376. PMID: 35923707; PMCID: PMC9339951.

Thank you for your valuable suggestions. We have carefully reviewed the recent publications highlighted by the reviewer and have incorporated relevant references into the revised version of the manuscript. Additionally, we have enriched the Results and Discussion sections with updated information to reflect the latest findings in the field (lines 152-189). We appreciate your insightful feedback, which has helped improve the quality of our work.

Reviewer 2 Report

Comments and Suggestions for Authors

The manuscript is well written, and the study is carefully performed—however, the sample size is only five subjects. The findings may not related to the disease, and I am concerned that some statements are overclaimed. The results and discussion need to be more elaborate. 

The statement in lines 61-62, "The most commonly used drug was enalapril," which is from three patients out of 5. Please rephrase it and avoid using "commonly" in this statement.

The results in the table need to be clarified.  

Typos:

Line 61 - should be "were recruited," not recluted

Line 65 - should be Antihistamines not AntiHistamines

Comments on the Quality of English Language

Please check the citation, format, and typos. 

Author Response

The manuscript is well written, and the study is carefully performed—however, the sample size is only five subjects. The findings may not related to the disease, and I am concerned that some statements are overclaimed. The results and discussion need to be more elaborate. 

Thank you for your thoughtful comments and constructive feedback. We acknowledge the limitations regarding the small sample size and have tone down the conclusions to address this point. We have also elaborated further on the results and discussion sections to provide a more thorough analysis and context for our findings. We have also made sure to clarify any statements that could be perceived as overclaimed, ensuring the conclusions are appropriately nuanced (lines 152-189).

The statement in lines 61-62, "The most commonly used drug was enalapril," which is from three patients out of 5. Please rephrase it and avoid using "commonly" in this statement.

Thank you for highlighting this point. We have revised the sentence accordingly (lines 70-71).

The results in the table need to be clarified.  

We have now provided more description and context in the results and discussion sections. We have also summarized all the information and presented in the table to ensure clarity and a more comprehensive understanding of the findings (lines 152-189).

Typos:

Line 61 - should be "were recruited," not recluted

Thank you very much for detecting this typo. It has been addressed in the revised version of this manuscript (line 70).

Line 65 - should be Antihistamines not AntiHistamines

Thank you very much for detecting this typo. It has been addressed in the revised version of this manuscript (line 74).

Reviewer 3 Report

Comments and Suggestions for Authors

Dear Authors,

I have reviewed your manuscript describing a pilot study of whole exome sequencing in ACEi-induced angioedema. Your work addresses an important clinical challenge and provides valuable preliminary genetic insights. Here are my detailed recommendations for strengthening the manuscript:

Introduction: It effectively establishes the clinical relevance of ACEi-induced angioedema. You could consider expanding on the clinical burden of ACEi-induced angioedema in current medical practice, including prevalence data and management challenges. This would help clinicians better appreciate the significance of genetic screening.

A more detailed explanation of how genetic variants might influence individual susceptibility to ACEi-induced angioedema. This would create a stronger foundation for your genetic investigation approach.

Methods: While your methodological approach is sound, several aspects need clarification for reproducibility:

Patient Selection: Expand on how the "life-threatening episode" was defined clinically. Include details about the specific criteria used to classify angioedema severity.

Provide information about any concurrent medications or comorbidities that might influence angioedema presentation.

Genetic Analysis: Include quality control metrics for the sequencing data, such as coverage depth statistics and variant calling parameters.

Explain the rationale behind your variant filtering strategy, particularly the choice of specific pathogenicity prediction tools.

Results: Your results present interesting genetic findings, but the presentation could be enhanced:

Consider adding a flow diagram showing the variant filtering process, from initial variants to final candidates.

Reorganize Table 2 to improve readability. You could consider splitting it into smaller, focused tables addressing different aspects (e.g., clinical characteristics, genetic findings).

For the variants identified, include more detailed analysis of their potential functional impact on protein structure or function.

Discussion: To maximize impact for clinicians, consider addressing:

The potential clinical implications of identifying these genetic variants for patient screening and risk assessment.

How these findings might influence the choice of antihypertensive medications in patients with similar genetic profiles.

A more detailed discussion of study limitations and specific recommendations for future larger-scale validation studies.

Several technical improvements would strengthen the manuscript:

Standardize the presentation of genetic variants throughout the text following HGVS nomenclature.

Include additional bioinformatics details in a supplementary methods section.

Consider adding a supplementary table with all identified variants of potential interest, not just the top candidates.

Style and Formatting For consistency with the Journal of Clinical Medicine:

Ensure all abbreviations are defined at first use.

Check reference formatting against the journal's style requirements.

To guide future studies, consider elaborating on:

Sample size calculations for a full-scale validation study.

Specific genetic variants that warrant priority investigation in larger cohorts.

Potential therapeutic implications of these genetic findings.

Author Response

Dear Authors,

I have reviewed your manuscript describing a pilot study of whole exome sequencing in ACEi-induced angioedema. Your work addresses an important clinical challenge and provides valuable preliminary genetic insights. Here are my detailed recommendations for strengthening the manuscript:

Introduction: It effectively establishes the clinical relevance of ACEi-induced angioedema. You could consider expanding on the clinical burden of ACEi-induced angioedema in current medical practice, including prevalence data and management challenges. This would help clinicians better appreciate the significance of genetic screening.

Thank you. We have expanded the introduction section following the suggestion. Specifically, we have provided prevalence data to highlight the extent of the condition and introduced the key challenges in its clinical management. These changes provide clinicians with a clearer understanding of the significance of ACEi-induced angioedema and underscore the importance of genetic screening in improving patient care (lines 43-53).

A more detailed explanation of how genetic variants might influence individual susceptibility to ACEi-induced angioedema. This would create a stronger foundation for your genetic investigation approach.

Thank you for your insightful comment. In response, we have expanded the description in the results and discussion sections, particularly regarding the genetic variants identified in the exome data of our patients. We have provided a more detailed explanation of how these variants might influence individual susceptibility to ACEi-induced angioedema, offering additional evidence supporting their potential role in the disease (lines 152-189).

Methods: While your methodological approach is sound, several aspects need clarification for reproducibility:

Patient Selection: Expand on how the "life-threatening episode" was defined clinically. Include details about the specific criteria used to classify angioedema severity.

Thank you for highlighting this point. We appreciate your careful review and have revised the concept "life-threatening episode" as suggested by the reviewer. We added dyspnea and oxygen saturation < 90% as criteria for severity

Provide information about any concurrent medications or comorbidities that might influence angioedema presentation.

As suggested, we have clarified that the patients do not present any pathology or chronic treatment that could influence the severity of symptoms related to AE-ACEi, ensuring that the genetic associations observed are not confounded by other prescribed treatments or comorbidities (lines 75-77).

Genetic Analysis: Include quality control metrics for the sequencing data, such as coverage depth statistics and variant calling parameters.

Thank you for your suggestion. We have revised table 2 to improve its readability and we have incorporated key sequencing metrics, including the total coverage depth and the read count supporting for each allele. Additionally, we have adapted the table to follow the HGVS nomenclature for genetic variants and have included the relevant bibliographic references associated with each prioritized variant accordingly (lines 225-227).

Explain the rationale behind your variant filtering strategy, particularly the choice of specific pathogenicity prediction tools.

Thank you for your valuable feedback. The aim of this approach was to capture all relevant genetic variants underlying AE-ACEi from the exome results. First, the search for hereditary angioedema (HAE) causal variants served to unravel undiagnosed borderline patients whose debut is caused by the initial administration of ACEi drugs. HADA was designed to identify previously reported robust causal variants of HAE. After excluding the presence of responsible genetic factors for HAE, we submit the exome data to Franklin to perform a broader genetic screening. We would like to note that, prior to selecting Franklin as the prioritization tool for this step, we have conducted comparisons of many publicly available tools for variant prioritization (Tosco-Herrera et al., Hum Mutat 2022; doi: 10.1002/humu.24459). This benchmark study highlighted the benefits to assist in identifying genetic variants associated with disease (https://github.com/genomicsITER/benchmark-germline-variants-prioritizers). Based on this evaluation, we found that Franklin's user-friendly interface, intuitive design, and high level of automation made it particularly suitable for our research and its expansion in the future with the inclusion of more affected patients. Additionally, its ability to filter by phenotypes, gene panels, and HPO terms aligns well with the needs of our study, especially when focusing on individual case analysis. We believe these features make it a strong candidate for genetic variant prioritization in our context.

Results: Your results present interesting genetic findings, but the presentation could be enhanced:

Consider adding a flow diagram showing the variant filtering process, from initial variants to final candidates.

We appreciate the reviewer’s suggestion. In the revised version of the manuscript, we have included a flow diagram illustrating the variant filtering process (Figure 1). This figure outlines the sequential steps followed in the exome sequencing of patients to prioritize novel candidate variants associated with AE-ACEi (lines 112-115).

Reorganize Table 2 to improve readability. You could consider splitting it into smaller, focused tables addressing different aspects (e.g., clinical characteristics, genetic findings).

Thank you for your valuable suggestion. We have modified the Table 2 to improve readability by enhancing its structure and presentation. Specifically, we have ensured a clearer organization of sequencing metrics and variant details. Additionally, we have considered splitting the information into more focused sections but decided to maintain a single table for coherence while optimizing its clarity. These adjustments are aimed to enhance the interpretability of the genetic findings while preserving all relevant data in a comprehensive format (lines 225-227).

For the variants identified, include more detailed analysis of their potential functional impact on protein structure or function.

Thank you for your valuable feedback. We have increased the description of genetic findings and provided more context in the results and discussion sections. We have also summarized all the information and presented in Table 2 to ensure clarity and a more comprehensive understanding of the findings (lines 152-189).

Discussion: To maximize impact for clinicians, consider addressing:

The potential clinical implications of identifying these genetic variants for patient screening and risk assessment.

Thank you for your suggestion. In the revised manuscript, we have expanded the discussion to highlight the potential clinical implications of identifying these genetic variants for patient screening and risk assessment. Specifically, we emphasize how the genetic findings could contribute to identifying individuals at higher risk of AE-ACEi, allowing to reach a personalized treatment approach for each patient included in our study and prevent these angioedema attacks restricting the use of ACEIs in favor of other antihypertensive treatments. To address this important aspect, we have revised the Results and Discussion section to explicitly highlight the potential clinical implications of our findings in the management of AE-ACEi patients (lines 203-219).

How these findings might influence the choice of antihypertensive medications in patients with similar genetic profiles.

Thank you for your valuable suggestion. These findings have the potential to be integrated into routine clinical practice in the future, contributing to the advancement of precision medicine. The identification of genetic variants associated with an increased risk of AE-ACEi could enable early screening of patients before initiating ACEi treatment. This approach would allow clinicians to personalize antihypertensive therapy by selecting alternative treatments for individuals with a genetic predisposition to angioedema, thereby minimizing the risk of adverse reactions. As genomic detection techniques become more accessible, incorporating genetic screening into clinical decision-making may enhance patient safety and treatment efficacy. To address this important aspect, we have revised the Results and Discussion section to explicitly highlight the potential clinical implications of our findings in the management of AE-ACEi patients (lines 203-219).

A more detailed discussion of study limitations and specific recommendations for future larger-scale validation studies.

We appreciate the reviewer’s insightful comment. We have expanded the discussion with the main limitations, especially exposing the small sample size and the lack of functional data to further validate the pathogenic potential of the identified variants. Additionally, we have included specific recommendations for future studies, underlining the need for larger cohorts and functional assays to confirm the role of these variants in the pathogenesis of ACEi-induced angioedema. These changes have been incorporated into the revised version of the manuscript (lines 191-219).

Several technical improvements would strengthen the manuscript:

Standardize the presentation of genetic variants throughout the text following HGVS nomenclature.

Thank you for your valuable suggestion. We have carefully revised the manuscript to ensure that all genetic variants are presented following the HGVS nomenclature. These changes have been incorporated into the revised version of the manuscript for consistency and clarity.

Include additional bioinformatics details in a supplementary methods section.

Thank you for your suggestion. We have carefully reviewed the bioinformatics details included in the manuscript and believe that the Materials and Methods section already provides a comprehensive description of the entire computational workflow used in this pilot study. This includes information about DNA extraction, sequencing protocols, data preprocessing, quality control steps, the variant calling procedure, filtering strategies to remove false positive and false negative results on variant calls, and the annotation methods to extract clinically relevant information of identified variation. Additionally, we have made the full bioinformatics pipeline publicly available at https://github.com/genomicsITER/benchmarking/tree/master/WES, where further details on the processing of exome data can be accessed. Given this, we consider that all necessary information is already provided in the revised version of the manuscript.

Consider adding a supplementary table with all identified variants of potential interest, not just the top candidates.

Thank you for your valuable suggestion. We have carefully considered the inclusion of a supplementary table with all identified variants. However, upon review, we believe that most of the additional variants do not show potential association with the disease under study. Including them may lead to confusion, as they are unlikely to contribute meaningful insights into the genetic underpinnings of ACEi-induced angioedema. Most of the prioritized variants using the filtering strategy detailed in the Materials and Methods section have been classified as benign, with no clear relation to metabolic pathways that could be relevant in the context of the disease. Therefore, we preferred to focus on the most relevant candidates to avoid unnecessary complexity in the presentation of the results.

Style and Formatting For consistency with the Journal of Clinical Medicine:

Ensure all abbreviations are defined at first use.

We appreciate the reviewer’s suggestion. We have carefully reviewed the manuscript to ensure that all abbreviations are defined at their first use and have made the necessary corrections in the revised version of the manuscript.

Check reference formatting against the journal's style requirements.

We appreciate the reviewer’s comment. We have carefully checked the reference formatting and ensured that it adheres to the journal's style requirements in the revised version of the manuscript.

To guide future studies, consider elaborating on:

Sample size calculations for a full-scale validation study.

We appreciate the reviewer’s suggestion. In the next steps of our study, we have planned to expand the cohort by incorporating additional AE-ACEi affected patients. This will allow us to perform more robust statistical analyses and improve the validation of the identified candidate variants. We acknowledge the importance of determining an appropriate sample size for a full-scale validation study and will take this into account in future research efforts. We have incorporated several changes in the revised version of the manuscript that clarify this point (lines 191-206).

Specific genetic variants that warrant priority investigation in larger cohorts.

We appreciate the reviewer’s comment. In this study, we have described several genetic variants in the ACE gene, which, to date, remains the primary genetic factor investigated due to its central role in the pathophysiology of ACEi-induced angioedema. Given its biological relevance, ACE has been the main candidate to explain the symptoms observed in affected individuals. However, our study aims to contribute to the broader search for genetic factors that may underlie the risk to this condition, potentially expanding the spectrum of variants and genes implicated in its pathogenesis. Further investigations in larger cohorts will be essential to validate these findings and identify additional genetic contributors. We have made several modifications in the updated manuscript to provide further clarification on this point (lines 165-189).

Potential therapeutic implications of these genetic findings.

We appreciate the reviewer’s insightful comment. While our study primarily focuses on identifying potential genetic variants associated with ACEi-induced angioedema (AE-ACEi), we agree that understanding the therapeutic implications of these genetic findings is crucial. The identification of genetic variants, particularly those in genes such as ACE, could help refine risk stratification in patients prone to AE-ACEi. In the future, genetic screening may enable personalized treatment plans, where patients with specific genetic profiles could be managed with tailored therapies to reduce the risk of angioedema induced by ACEi intake. This approach could also guide clinicians in a high-risk decision-making situation about the treatment for the patients and improve the overall management of AE-ACEi. We have incorporated several changes in the revised version of the manuscript that clarify this point (lines 206-219).

Round 2

Reviewer 1 Report

Comments and Suggestions for Authors

Dear Author, After this improvement, the paper is suitable for publication.